# Gene flow contributes to diversification of the major fungal pathogen *Candida albicans*

Jeanne Ropars [1,2], Corinne Maufrais [1,3], Dorothée Diogo[1], Marina Marcet-Houben[1,4,5], Aurélie Perin[1], Natacha Sertour[1], Kevin Mosca[1], Emmanuelle Permal[1], Guillaume Laval[3,6], Christiane Bouchier[7], Laurence Ma[7], Katja Schwartz[8], Kerstin Voelz [9], Robin C. May [9], Julie Poulain[10,11,12], Christophe Battail[10], Patrick Wincker[10,11,12], Andrew M. Borman [13], Anuradha Chowdhary[14], Shangrong Fan[15], Soo Hyun Kim [16], Patrice Le Pape[17], Orazio Romeo [18,19], Jong Hee Shin [16], Toni Gabaldon[4,5,20], Gavin Sherlock [8], Marie-Elisabeth Bougnoux[1,21,22] & Christophe d'Enfert [1]

Elucidating population structure and levels of genetic diversity and recombination is necessary to understand the evolution and adaptation of species. *Candida albicans* is the second most frequent agent of human fungal infections worldwide, causing high-mortality rates. Here we present the genomic sequences of 182 *C. albicans* isolates collected worldwide, including commensal isolates, as well as ones responsible for superficial and invasive infections, constituting the largest dataset to date for this major fungal pathogen. Although, *C. albicans* shows a predominantly clonal population structure, we find evidence of gene flow between previously known and newly identified genetic clusters, supporting the occurrence of (para) sexuality in nature. A highly clonal lineage, which experimentally shows reduced fitness, has undergone pseudogenization in genes required for virulence and morphogenesis, which may explain its niche restriction. *Candida albicans* thus takes advantage of both clonality and gene flow to diversify.

[1] Department of Mycology, Fungal Biology and Pathogenicity Unit, Institut Pasteur, INRA, 75015 Paris, France. [2] Ecologie Systematique et Evolution, CNRS, Univ. Paris Sud, AgroParisTech, Université Paris Saclay, 91405 Orsay cedex, France. [3] Center for Bioinformatics, BioStatistics and Integrative Biology (C3BI), USR 3756 IP CNRS, Institut Pasteur, 75015 Paris, France. [4] Centre for Genomic Regulation (CRG), The Barcelona Institute for Science and Technology, 08003 Barcelona, Spain. [5] Universitat Pompeu Fabra (UPF), 08002 Barcelona, Spain. [6] Department of Genomes and Genetics, Human Evolutionary Genetics Unit, UMR 2000 CNRS, Institut Pasteur, 75015 Paris, France. [7] Biomics Pole, CITECH, Institut Pasteur, 75015 Paris, France. [8] Department of Genetics, Stanford University Medical School, Stanford, CA 94305-5120, USA. [9] School of Biosciences and Institute of Microbiology and Infection, University of Birmingham, Birmingham B15 2TT, UK. [10] CEA, Genoscope, Institut de biologie François Jacob, 91000 Evry, France. [11] CNRS UMR 8030, 91000 Evry, France. [12] Univ. Evry, Univ. Paris-Saclay, 91000 Evry, France. [13] UK National Mycology Reference Laboratory, Public Health England, Bristol BS2 8EL, UK. [14] Department of Medical Mycology, Vallabhbhai Patel Chest Institute, University of Delhi, Dehli 110007, India. [15] Department of Obstetrics and Gynecology, Peking University Shenzhen Hospital, PR Guangdong Sheng 518036, China. [16] Department of Laboratory Medicine, Chonnam National University Medical School, Gwangju 61469, South Korea. [17] EA1155 – IICiMed, Institut de Recherche en Santé 2, Université de Nantes, 44200 Nantes, France. [18] Department of Chemical, Biological, Pharmaceutical and Environmental Sciences, University of Messina, 98166 Messina, ME, Italy. [19] IRCCS – Centro Neurolesi Bonino-Pulejo, 98124 Messina, Italy. [20] ICREA, 08010 Barcelona, Spain. [21] Unité de Parasitologie-Mycologie, Service de Microbiologie clinique, Hôpital Necker-Enfants-Malades, Assistance Publique des Hôpitaux de Paris (APHP), 75015 Paris, France. [22] Université Paris Descartes, Sorbonne Paris-Cité, 75006 Paris, France. Correspondence and requests for materials should be addressed to C.d'E. (email: christophe.denfert@pasteur.fr)

Elucidating population subdivision and levels of genetic diversity and recombination are necessary steps for understanding the evolution and adaptation of species. It can reveal allopatric differentiation, host adaptation or other types of local adaptation as consequences of reduction of gene flow promoting genetic drift and natural selection[1]. For example, the causal agent of the white-nose syndrome in bats, the killer of millions of bats in North America since its discovery in 2006, was shown to be a single clone of the fungus *Pseudogymnoascus destructans*[2]. Indeed, population genetics has revealed the occurrence of only spontaneous mutations in this pathogen with no indication of recombination. Thus, studying population genetics of pathogens has a clear applied importance toward the understanding of disease emergence through adaptation or drug resistance.

It is estimated that 5 million fungal species exist, yet only a few hundred are known to cause disease in humans[3]. Among the latter, *Candida albicans* belongs to one of the four genera causing high-mortality rates in humans and is the second most frequent agent of fungal infection worldwide[4]. While *C. albicans* is part of the normal human intestinal microbiota, it also causes mucosal diseases in healthy individuals, as well as deep-seated opportunistic infections in hosts with decreased defenses (e.g., immunocompromised individuals, patients who have endured invasive clinical procedures or have experienced major trauma).

*Candida albicans* is a predominantly diploid species, possessing a parasexual cycle[5,6] which differs from a conventional sexual cycle by the lack of meiosis. In brief, the parasexual cycle of *C. albicans* involves (i) the fusion of two diploid cells carrying opposite mating types (syngamy), followed by (ii) nuclear fusion (karyogamy) and (iii) concerted chromosome loss to return to the diploid state, replacing conventional meiosis. Despite the absence of meiosis, the parasexual cycle of *C. albicans* allows chromosome shuffling and recombination events by means of gene conversion and mitotic recombination, likely contributing to the genetic and phenotypic diversity in this species[5,6]. However, the evidence of the importance of parasexuality in nature is lacking, as genetic analyses have identified predominantly clonal populations (or genetic clusters, also known as clades) in *C. albicans*[7–13].

Previous studies have reported significant genetic diversity across *C. albicans* clinical isolates using either a mildy-repetitive DNA fingerprinting probe[8–10,14], several genes as probes[11] or multilocus sequence typing (MLST), the latter having been widely used to type *C. albicans* isolates in the past 15 years[12,13,15,16]. To date, 18 genetic clusters have been identified using MLST (numbered 1–18). These clusters may have geographic origins and they display some phenotypic specificities (reviewed in ref.[16]). In 2015, the comparative genomic analysis of 21 clinical isolates, which had been previously assigned to existing MLST clades, recapitulated relationships between isolates, and the authors reported the discovery of extensive variation between these 21 isolates, including single nucleotide polymorphisms (SNPs) and frequent whole or partial chromosomal aneuploidies[17]. Strikingly, these isolates showed a high frequency of homozygosity at the genomic region controlling fungal compatibility (also called the mating-type locus in fungi, 12/21 isolates, 57%), which is in contrast with what was previously reported (110/1294 strains, 8.5%[12]). This may reflect antifungal exposure of these clinical isolates, which also likely explains their high frequency of aneuploidies[18].

In this work, we sequenced the genomes of 182 *C. albicans* isolates collected worldwide, an order of magnitude more isolates than has been considered previously in population genomic studies of *C. albicans*. Our dataset contains commensal isolates, as well as ones responsible for superficial and invasive infections, and also includes the previously sequenced laboratory strain SC5314[19] and representatives from all major clusters previously defined by MLST[12,13,15]. While *C. albicans* shows a predominantly clonal population structure, our analyses show evidence of introgressions (or admixture) in two newly identified genetic clusters, supporting the occurrence of (para)sexuality in nature. Importantly, a highly clonal lineage, which experimentally showed reduced fitness, has undergone pseudogenization in genes required for virulence and morphogenesis, which may explain its niche restriction.

## Results and Discussion

**Aneuploidies are rare and loss of heterozygosity (LOH) frequent.** The Illumina® technology was used to deep sequence 182 *C. albicans* isolates, including the previously sequenced laboratory strain SC5314[19] and representatives from all major clusters previously defined by MLST[12,13,15] (Supplementary Data 1). Across all strains, we identified a total of 589,255 SNPs (Supplementary Data 2; see Methods section for details). On a broad scale, we observed segmental aneuploidies in eight strains (Supplementary Data 1) and whole-chromosome aneuploidies in ten (Supplementary Data 1, Supplementary Fig. 1), suggesting that the high rate of aneuploidies previously described[17] is an exception rather than the rule, likely due to antifungal treatments[20]. Consistent with previous surveys[12,21], only four sequenced strains (2.2%) showed a homozygous mating-type locus.

Phenotypic diversity in *C. albicans* can arise rapidly through LOH, spanning whole chromosomes or shorter contiguous chromosome segments[18,22]. We detected numerous LOH events across our 182 isolates. Some were ancient events that had arisen before cluster expansion (Fig. 1, examples highlighted by black dotted boxes) and others were more recent and strain specific (Fig. 1, horizontal white stripes specific to a single strain). Long range LOH events were predominantly the consequence of mitotic crossovers or break-induced replication events while events of whole chromosome loss were rare (Fig. 1). Strains belonging to clade 13 showed a unique pattern with lower heterozygosity (Fig. 1 and see below). Analysis of genome-wide variation revealed that each of the 182 isolates on average contain 65,629 heterozygous SNPs (1 heterozygous SNP every 204 bp, Supplementary Data 1) and 14,189 heterozygous insertion-deletion events (indels), which is in agreement with previous genomic analysis[17].

**A predominantly clonal population structure of *C. albicans*.** Maximum-likelihood phylogenetic analysis based on 264,999 highly confident SNPs (SNPs across the 182 isolates with no missing data for all strains, Supplementary Data 3) yielded a tree showing 17 distinct genetic clusters, including 12 previously found using multilocus sequencing typing (MLST)[12,13,15] and five new ones (Fig. 2a). A majority of the isolates belonged to clusters 1 ($n = 40$), 2 ($n = 15$), 3 ($n = 11$), 4 ($n = 27$), 11 ($n = 10$), and 13 ($n = 35$) (Fig. 2a). Fixation indices ($F_{ST}$) further confirmed a high-genetic differentiation between clusters (mean $F_{ST} = 0.83$; Supplementary Table 1). Ten isolates could not be assigned to any cluster, likely because they belong to undersampled or rarer clusters. Comparing this tree with those obtained using indels and transposable elements showed the same delimitations of clusters, consistent with a predominantly clonal population structure of *C. albicans*[7,12] (Fig. 2a and Supplementary Figs. 2, 3). In addition, we found an excess of heterozygous SNPs within clusters when analyzing SNPs specific to this cluster (Meselson effect[23], Supplementary Fig. 4), further confirming the clonal expansion of this human fungal pathogen.

We performed a neighbor-net analysis using the network approach to visualize possible recombination events within and

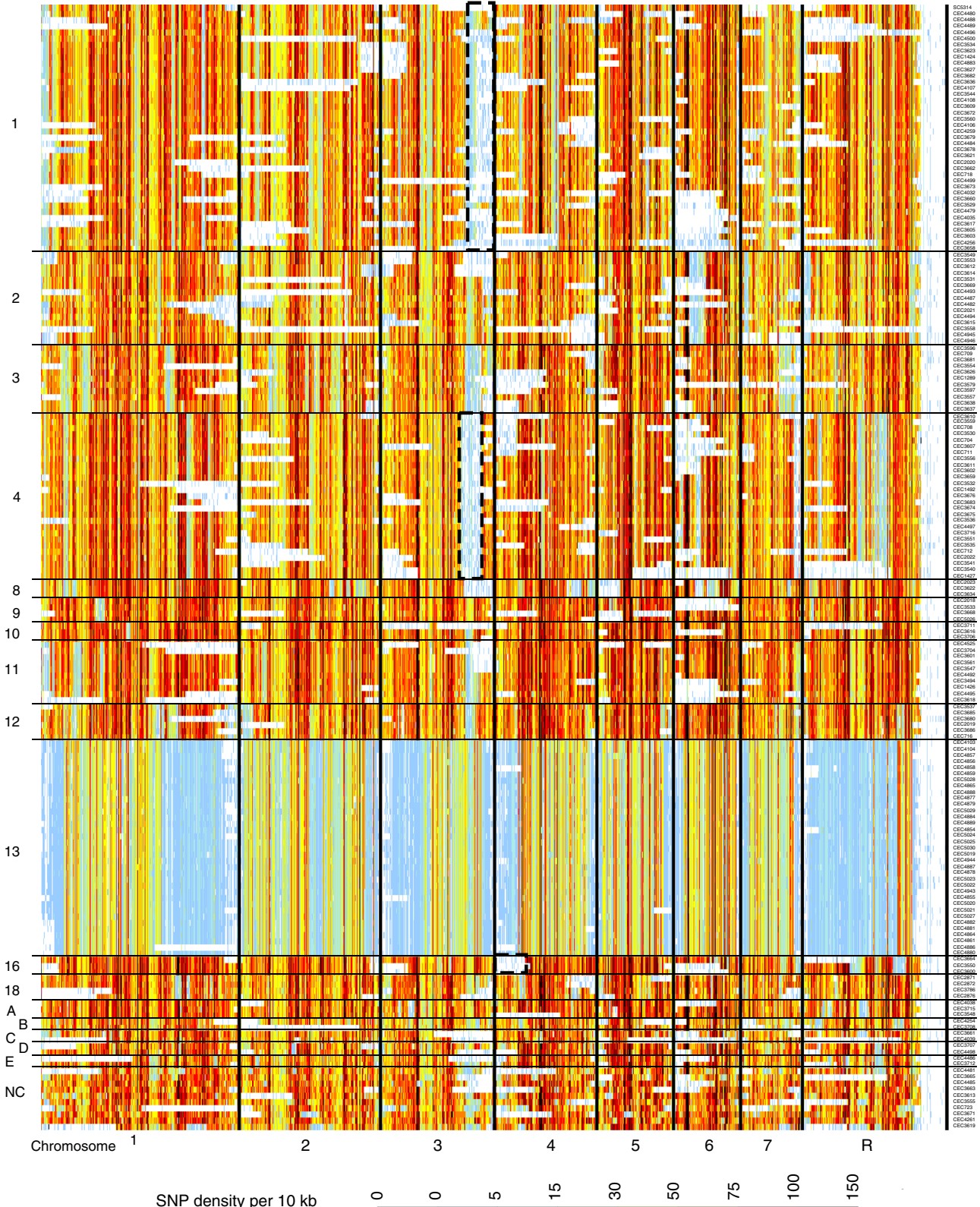

**Fig. 1** Density of heterozygous SNPs in 182 *C. albicans* isolates, in 10 kb windows. Each row represents a strain. Strains are ordered according to their cluster assignation. Thick vertical black lines delimit chromosomes (from 1 to 7 and R). Dotted black boxes highlight examples of ancestral LOH shared by all isolates of a cluster. Horizontal white stripes are indicative of recent LOH events. The scale bar represents density of heterozygous SNPs per 10 kb window, from a low density in light blue (white for 0) to a high density in dark red

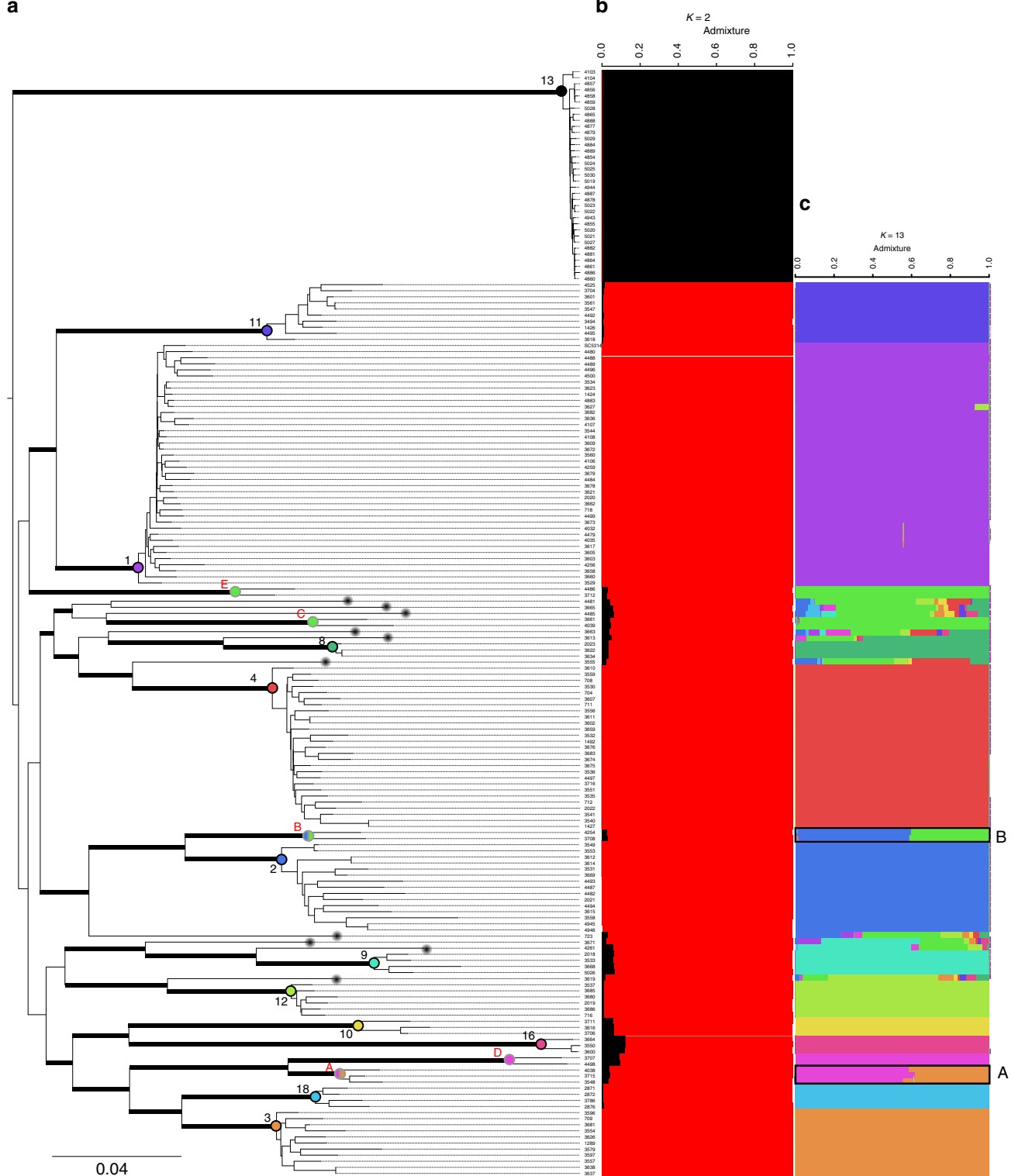

**Fig. 2** Phylogenetic relationships and population structure of *Candida albicans*. **a** Maximum likelihood tree showing phylogenetic relationships between the 182 isolates used in this study; Thick bars represent bootstrap supports >95% (bootstrap analysis of 1000 resampled datasets); branch lengths are shown and the scale bar represents 0.04 substitutions per site. We used the midpoint rooting method to root the tree. Clusters already described in previous studies using MLST data are written in black (from 1 to 18, incomplete due to sampling) and new clusters described in this study are written in red and named with letters from A to E. Black dots at the end of some branches (10 in total) pinpoint strains which could not be assigned to any cluster. **b**, **c** Population structure of *C. albicans* at **b** $K = 2$ and **c** $K = 13$. The structure has been inferred using NgsAdmix. Each line represents a strain, as in the ML tree (**a**) and colored bars represent their coefficients of membership in the various gene pools based on SNP data

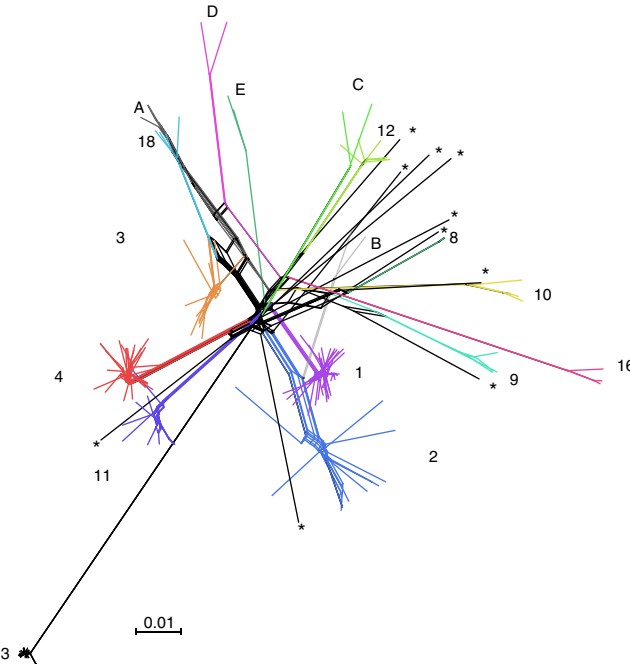

**Fig. 3** Neighbor-net of *Candida albicans*. Neighbor-net analysis for 182 isolates collected worldwide based on SNP data. Reticulation indicates likely occurrence of recombination. Branch lengths are shown and the scale bar represents 0.01 substitutions per site. Asterisks pinpoint strains which could not be assigned to any genetic clusters

between lineages. Although this analysis further confirmed the predominant clonality of *C. albicans* (Fig. 3), it also revealed some footprints of recombination indicated by reticulated patterns. This was also confirmed by Pairwise Homoplasy Tests (phi tests) conducted on ORFs using 1000 random permutations of the SNP positions, based on the expectation that sites are exchangeable without recombination[24] (*p*-value = 0.00; 877/6590 features, Supplementary Data 4).

**Footprints of admixture in two *C. albicans* genetic clusters.** Detection of recombination (Fig. 3) led us to further investigate possible footprints of admixture in our dataset. We inferred individual ancestry based on genotype likelihoods from realigned reads by assuming a known number of admixing populations ranging from 2 to 16, using the software NgsAdmix[25]. At *K* = 2, two well-delimited populations were found, separating strains from cluster 13 (in black, Fig. 2b) from all other strains. Because cluster 13 isolates showed different SNP patterns compared to other strains (Fig. 1 and see below), we reanalyzed the data after removing this cluster. At several *K* (from 8 to 16), two genetic clusters always appeared admixed, namely clusters A and B (Fig. 2c). Genomic scans of statistics designed for measuring population differentiation, i.e., $F_{ST}$ and df (number of fixed differences between each pair of clusters), were performed to localize regions of introgression and to assess their origin (Fig. 4). Cluster A showed footprints of introgression from three different clusters (clusters 3, D and 18, Fig. 4a and Supplementary Fig. 5) while cluster B only showed footprints of introgression from cluster 2 (Fig. 4b and Supplementary Fig. 6). Neighbor-net analyses only including newly identified admixed genetic clusters and their supposed ancestors (clusters A, 3, D and 18, Fig. 4c; clusters B and 2, Fig. 4d) using isolates of clusters 1 or 4 as outgroups, confirmed the presence of recombination as shown by reticulation between these populations.

In *C. albicans*, a parasexual cycle has been described, differing from the conventional sexual cycle by the absence of meiosis, which is replaced by concerted chromosome loss after nuclear fusion to return to the diploid state[5,6,26]. However, no evidence of introgression has been described in nature. Here we report clear evidence that new clusters can arise from introgression events in this widespread human pathogen. This parallels recent findings in another putatively asexual opportunistic human pathogen *Candida glabrata*[27], and suggests that gene flow in *Candida* pathogens may be more common than previously thought.

Genetic differentiation occurs when gene flow is prevented, due to reproductive barriers or asexuality. Genes involved in mating were shown to be under purifying selection in *C. albicans*[28], suggesting that they are still functional. Accordingly, mating has been induced between various *C. albicans* isolates of opposite mating-types as well as between *C. albicans* and its close relative *C. dubliniensis*, in the laboratory in vitro and in vivo, leading to the formation of tetraploids[29–32]. Return to the diploid state has also been observed, and involved random chromosome loss rather than meiosis[5]. While these data suggest that genetic differentiation in the *C. albicans* species does not impose prezygotic barriers, these experiments did not systematically address possible genetic incompatibilities between genetic clusters. In particular, they did not assess the existence of postzygotic reproductive isolation, i.e., non-viability or sterility of hybrids. Thus, a thorough investigation of reproductive isolation between different genetic clusters in *C. albicans* is still lacking. Importantly, the two newly identified genetic clusters showing footprints of admixture were previously unknown. We believe that our resource of 182 genome-sequenced isolates will be invaluable to address this key question.

**A highly clonal lineage with reduced fitness and pseudogenes.** Cluster 13 showed very short branches in the phylogenetic tree (Fig. 2) and no reticulation in the neighbor-net analysis (Fig. 3). Furthermore, cluster 13 isolates showed different SNP patterns compared to strains from other clusters (Fig. 1). Indeed, their index of nucleotide diversity π was much lower than that for other clusters ($\pi_{cluster13}$ = 0.14 versus mean $\pi_{clusters1,2,3,4,11}$ = 0.36 in the five most represented clusters, Supplementary Table 2). Cluster 13 also showed lower polymorphism than other clusters (12,310 polymorphic sites in cluster 13 in contrast to 30,334 on average in the five other most represented clusters, Supplementary Table 3). Strikingly, the number of heterozygous SNPs was much lower in this cluster (average: 39,616), as compared to others (average: 69,740; Fig. 1, Supplementary Fig. 7). Cluster 13 has been proposed to be ranked as a new species, named *C. africana*[33], because the first isolated strains were from Africa, and were morphologically and physiologically different from other strains of *C. albicans*[33–38] (e.g., slower growth, inability to produce chlamydospores and to assimilate trehalose or amino sugars). All isolates of cluster 13 were collected from the genital tract and showed lower virulence in animal models of *Candida* infections[35]. Indeed, strains from cluster 13 studied here showed reduced fitness, i.e., slower growth rates on different media at different temperatures than strains from other clusters (Supplementary Data 5). We measured fitness in vaginal simulative medium (VSM), saliva simulative medium (SSM), and rich medium (YPD), and confirmed that strains from cluster 13 are not more fit in the genital niche than other strains of *C. albicans* (Supplementary Data 5, Fig. 5), instead suggesting a niche restriction due to a defect in fitness in other parts of the human body rather than specific adaptation to the genital niche.

Because cluster 13 isolates showed reduced fitness and a decrease in virulence compared to other clusters of *C. albicans*[35],

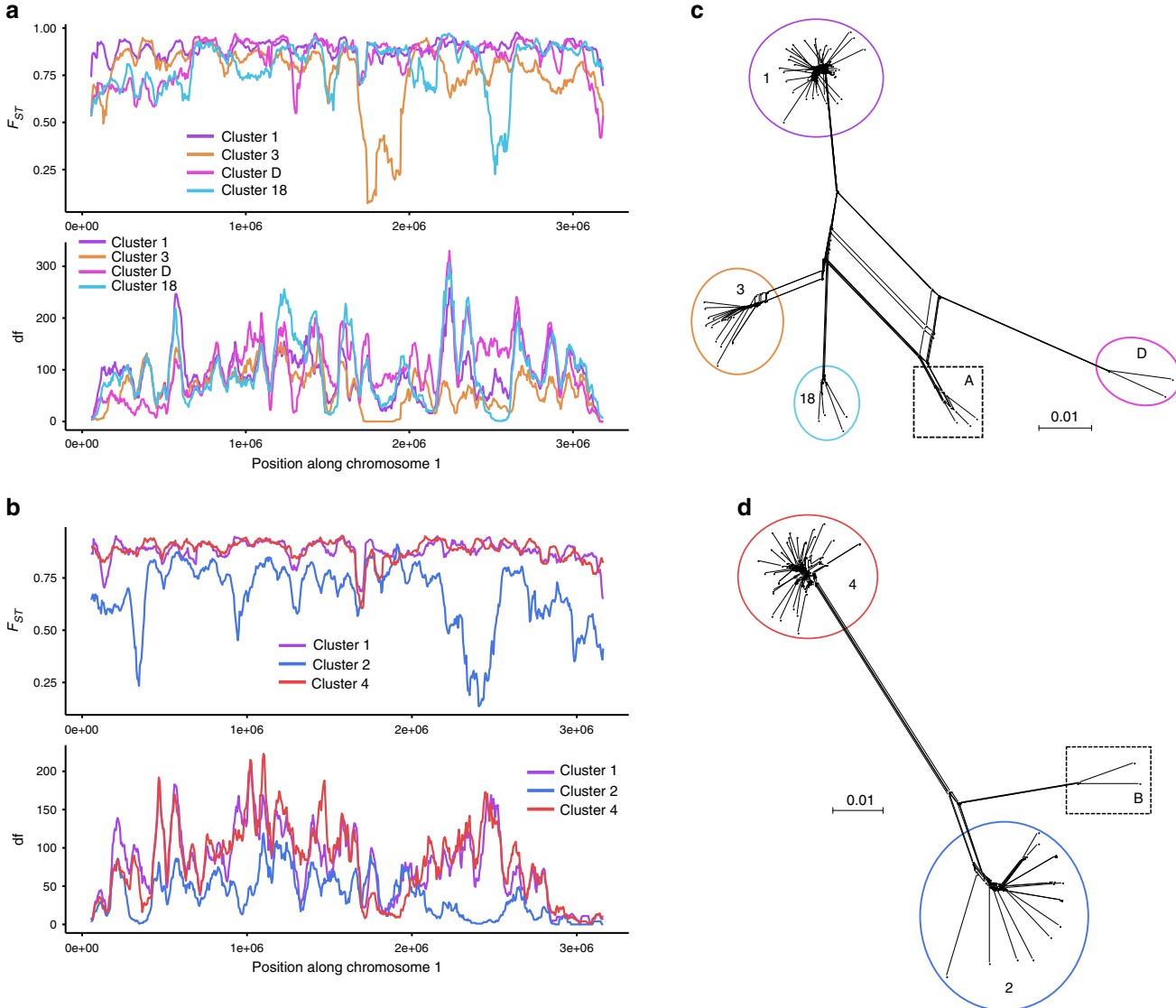

**Fig. 4** Evidence of admixture in clusters A and B. **a** and **b** Genomic scans of $F_{ST}$ (fixation index, an index measuring the differentiation between populations) and df (the number of fixed differences between populations) in sliding windows of 50 kb using a window step of 5000 bp. Predicted regions of admixture correspond to those that show a drop in both $F_{ST}$ and df. **a** Comparison of cluster A to clusters 1, 3, D and 18 along chromosome 1; **b** comparison of cluster B to cluster 1 and 2 along chromosome 1; **c**, **d** neighbor-net analyses based on SNP data including **c** clusters A, 3, D, 18 and 1 and **d** clusters 2, B and 1; branch lengths are shown in **c** and **d** and the scale bar represents 0.01 substitutions per site

we hypothesized that some genes important for virulence and growth may be missing and/or disrupted in this cluster. To test our hypothesis, we investigated the presence of homozygous premature stop codons due to nonsense mutations in the 6179 predicted ORFs of *C. albicans*. While no premature stop codons were fixed in and specific to clusters 1, 2, and 4, two such stop codons were detected in cluster 3 and one in cluster 11; however, mutations in these genes have not been reported as impacting survival or virulence in *C. albicans*. By contrast, 39 ORFs showed premature stop codons that were fixed in, and specific to cluster 13 (Supplementary Data 6, five were confirmed by Sanger sequencing). These ORFs included genes encoding transcription factors required for fitness in systemic infection and proper regulation of morphogenesis, such as *SFL1*[39] and *ZCF29*[40].

High rates of clonal reproduction have been both theoretically and empirically reported to increase the effective number of alleles and heterozygosity in a population[41–43]. In *C. albicans*, strain-specific recessive deleterious/lethal alleles have been

identified and shown to limit LOH[44]. Cluster 13 isolates however have much lower heterozygosity compared to other strains of *C. albicans* (Fig. 1). This may reflect a combination of massive ancestral LOH events and clonal reproduction in this cluster, with fixation of several deleterious alleles, affecting the overall fitness of these strains and leading to its niche restriction. Notably, the closest relative of *C. albicans*, namely *Candida dubliniensis*, also shows lower heterozygosity[45]. It has been reported that *C. dubliniensis* is less virulent and has lower fitness compared to *C. albicans*[46–49], and that its genome harbors numerous pseudo-genes and a lower level of genetic diversity. In the CTG clade of *Saccharomycotina* to which *C. albicans* and *C. dubliniensis* belong, more distantly related diploid species show high levels of heterozygosity similar to (or even higher than) those observed for non-cluster 13 isolates of *C. albicans*. These species have been shown to result from hybridization events[50–52]. Cluster 13 isolates of *C. albicans* showed ancient LOH in telomere-proximal regions, suggesting these events have occurred by break-induced

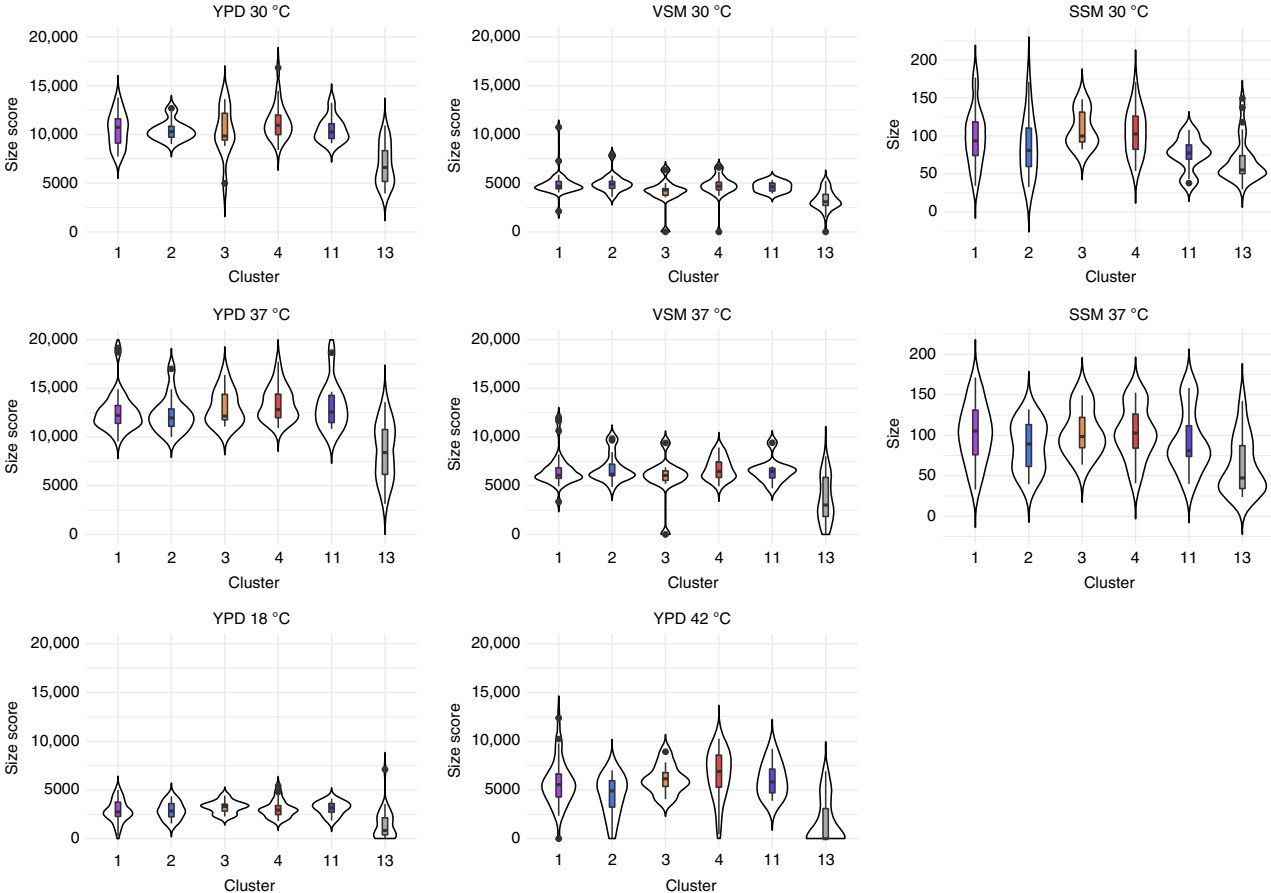

**Fig. 5** Strains of cluster 13 grow less well than other strains of *C. albicans* in different media at different temperatures. Violin plots of growth size scores performed using Iris by grouping isolates by clusters (only the most represented clusters are shown, i.e., those with >10 individuals: clusters 1, 2, 3, 4, 11 and 13) on vaginal-simulative medium (VSM) and YPD; For saliva simulative medium (SSM), we calculated colony size as mean of two perpendicular measures of diameters using ImageJ. Horizontal lines in the boxplots represent the median, vertical ones the length between upper and lower quartiles. Dots represent the outlier values. Some strains showed very poor growth on VSM, preventing measurement of colony size. These strains were not included in the figure. All experiments were performed in duplicate. In all conditions (different media and different temperatures), cluster 13 isolates are significantly less fit than other strains of *C. albicans* (ANOVA analyses followed by post-hoc comparisons using TukeyHSD tests (see Supplementary Data 5)). VSM: vaginal-simulative medium, SSM: saliva-simulative medium, YPD: yeast peptone dextrose medium

replication (BIR) and/or mitotic recombination, as regions around centromeres showed higher heterozygosity (Fig. 1 and Supplementary Fig. 7). This pattern of heterozygosity is different from what we observed in other *C. albicans* isolates, in which heterozygosity was steady along chromosomes, except when recent LOH events were observed (Fig. 1 and Supplementary Fig. 7). The most parsimonious ancestral state is thus a high level of heterozygosity, followed by two independent events (homoplasy) of massive losses of heterozygosity and accumulations of deleterious alleles in *C. dubliniensis* and isolates of cluster 13 of *C. albicans*.

In conclusion, our population genomic analyses shed light on the processes of divergence—namely (para)sexuality (as evidenced by gene flow) and clonality—in the most widespread opportunistic human fungal pathogen, *C. albicans*. We believe that the availability of 182 genome sequences of *C.albicans* isolates constitutes an invaluable genetic resource for the scientific community, not only for specialists of this species to better understand the biology of *C. albicans*, but also for evolutionary biologists to comprehend disease emergence.

## Methods
**Sampling**. A total of 182 isolates of *Candida albicans* were collected previously from different continents and origins (Supplementary Data 1).

**Production of whole-genome sequencing data**. Genomic DNA was extracted from the colonies using the phenol chloroform protocol previously described in ref. [15] or the QiaAmp DNA Mini Kit (Qiagen). The genomes were sequenced either at the Biomics Pole—Genomic Platform of Institut Pasteur, the Department of Genetics at Stanford University or the Sequencing facility of the University of Exeter (see Supplementary Data 1 for details) using the Illumina sequencing technology. Paired-end reads of 100–125 bp were obtained. Reads have been deposited at the NCBI Sequence Read Archive under BioProject ID PRJNA432884.

Each set of paired-end reads was mapped against the *C. albicans* reference genome SC5314 haplotype A or haplotype B[53] downloaded from the Candida Genome Database[54] (version A22 06-m01) using the Burrows–Wheeler Alignment tool, BWA version 0.7.7[55], with the BWA-MEM algorithm, specifically designed for sequences ranging from 70 bp to 1 Mb and recommended for high-quality queries. SAMtools version 1.2[56] and Picard tools version 1.94 (http://broadinstitute.github.io/picard) were then used to filter, sort and convert SAM files.

SNPs were called using Genome Analysis Toolkit version 3.1–1[57–59], according to the GATK Best Practices. SNPs and indels were filtered using these following parameters: VariantFiltration, QD < 2.0, LowQD, ReadPosRankSum < −8.0, LowRankSum, FS > 60.0, HightFS, MQRankSum < −12.5, MQRankSum, MQ < 40.0, LowMQ, HaplotypeScore > 13.0, HaploScore. Coverages were also calculated using the Genome Analysis Toolkit.

We created two tables encompassing all 182 isolates from VCF files using custom scripts. One encompassed 264,999 confident SNPs across the 182 isolates containing no missing data. Besides passing GATK's filters, we also checked for read depth (it had to be between 0.5 and 1.5 of the mean genome coverage), heterozygous positions should have an allelic ratio of number of alternative allele reads/total number of reads comprised between 15 and 85% and homozygous positions should have an allelic ratio of number of alternative allele reads/total number of reads >98% (Supplementary Data 2). The second table encompassed

589,255 SNPs where some of the new filters described above could be not respected and we created a code to have information of which filter did not pass:—for wrong allelic ratio of reference/alternative allele for heterozygous positions,++ for wrong allelic ratio of reference/alternative allele for homozygous positions, ## for a read depth not between 0.5 and 1.5 of the mean genome coverage; some positions could have several filters which did not pass: a combination of—and ## gave && and ++and ## gave ** (Supplementary Data 3).

**Phylogenetic analyses and distance trees**. RAxML[60] was used to infer phylogenetic relationships between the 182 isolates using the dataset of 264,999 confident SNPs using 1000 bootstraps replicates. As our dataset does not include any outgroup, we used the midpoint rooting method to root our tree, in which the root is set at the midpoint between the two most divergent isolates. We also created neighbor-joining trees using insertion/deletion events by coding no indel as 0, heterozygous indel as 1 and homozygous indel as 2, using the R package[61] ape[62]. The distance matrix was calculated by counting the number of differences.

**Genetic structure**. We used the dataset of 264,999 confident SNPs to infer the finer population structure within *C. albicans*. We performed NgsAdmix from the ANGSD package[63] to look for admixture in our dataset, from $K = 2$ to $K = 16$. After $K = 13$, clusters with the highest number of isolates, i.e., clusters 1 ($n = 40$) and 4 ($n = 27$), were split into sub-clusters, which likely reflect a problem of number of isolates within clusters rather than a biological meaning.

**Statistics of population genetics**. Nucleotide diversity ($\pi$) using VCFtools[64] with the—site-pi option was computed within each cluster. We also used ANGSD[63] to measure differentiation between populations (weighted $F_{ST}$)[65]. Three measures of divergence, $F_{ST}$ (using ANGSD) and df, the number of fixed differences between populations were computed using custom scripts. This was done along sliding windows of 50 kb using steps of 5000 bp. Plots were done using the R package ggplot2[66].

Linkage disequilibrium (LD) was computed as $r^2$, the coefficient of correlation between a pair of SNPs, with PLINK version 1.07[67], excluding SNPs with minor allele frequency lower than 0.05. LD ($r^2$) was calculated for each cluster with >10 individuals (clusters 1, 2, 3, 4, 11, and 13) for each chromosome. Mean $r^2$-values for each cluster for each chromosome were plotted using the R package ggplot2[66].

**Neighbor-net analyses**. We used the R package phangorn[68] for performing neighbor-net analyses.

**Hierarchical clustering based on the coverage of repeats**. To confirm the predominantly clonal propagation of *C. albicans*, we calculated the sequencing depth of 121 features annotated as "long_terminal_repeat", "retrotransposon" and "repeat_regions"[54]. For each feature, we normalized by the sequencing depth of the corresponding chromosome to remove the impact of potential aneuploidies. The clustering was generated by Cluster 3.0[69] using hierarchical clustering (complete linkage clustering) and the spearman rank correlation for measuring non-parametric distance, and visualized with java treeview[70] by converting values in log2 scale.

**Sequencing depth by bins of 1 kb**. To identify aneuploid chromosomes in the 182 strains, we calculated average sequencing depth on the eight chromosomes for each strain. Sequencing depth obtained for each bin of 1 kb on each chromosome was multiplied by the ploidy of the strain as defined from FACS analysis, divided by the genome sequencing depth and converted to log2 values. These values were then corrected through division by the median of all values obtained for chromosomes that had an average sequencing depth that did not deviate by >20% from the average sequencing depth of the whole genome. This allows the median of values obtained for a diploid chromosome to be ~1; whereas, the median of values obtained for a triploid chromosome is ~1.58 and the median of values obtained for a tetraploid chromosome is ~2. In the absence of this correction, values of diploid chromosomes are underestimated if the strain harbors triploid or tetraploid chromosomes. Averages of the normalized value obtained for each of the eight chromosomes in each strain were calculated and used to generate a heatmap.

**Flow cytometry analysis**. For each of the 182 strains, cells from the frozen collection (temperature: −80 °C) were grown in tubes for 36 h at 30 °C under agitation in 3 mL of YPD medium (1% yeast extract, 2% peptone, 2% dextrose). We then collected 1 mL of culture (about $1 \times 10^7$ cells/mL) in 2 mL Eppendorf tubes; cells were collected by centrifuging (5 min at 3500 r.p.m.) and resuspended in 300 μL of sterile water. We slowly added 700 μL of pure ethanol, repeatedly inverted tubes and incubated overnight at 4 °C. After centrifuging 5 min at 3500 r.p.m., cells were washed once with 1 mL of sterile water, resuspended in 0.5 mL of RNase solution (40 μg/mL; Thermo Fisher) and incubated for 4 h at 37 °C. Then, cells were collected by centrifuging (5 min at 3500 r.p.m.) and resuspended in 0.5 mL of 50 mM Tris-HCl (pH 8.0). 50 μL of suspension were transferred in hemolysis tubes with 0.5 mL of SYTOX Green (Invitrogen) staining solution (1 μM SYTOX Green in 50 mM Tris-HCl buffer, pH 8.0). Finally, samples (60,000 cells) were analyzed using a

MACSQuant (Miltenyi) flow cytometer, with a 488 nm laser to excite SYTOX Green and a bandpass filter 500–550 nm to detect fluorescence.

**Growth phenotypes on solid media**. The 182 isolates were split into three plates of 96, with the reference SC5314 present on each plate. Pre-cultures in deep wells from frozen cultures at −80 °C were realized at 30 °C for 36–48 h by taking 10 μL in a final volume of 500 μL in YPD liquid medium. Optical density was set to 1 for each isolate. We used the ROTOR from Singer Instruments to inoculate our 96 colonies at once, on solid media. All experiments were performed in duplicate. We inoculated strains on YPD medium and let them grow for 3 days into chambers at 18, 30, 37 and 42 °C. Pictures of plates were taken using the PhenoBooth from Singer Instruments at high quality (4128 × 3096). We also inoculated all strains included in this study on vaginal simulative medium (VSM), saliva simulative medium (SSM) and YPD media (YPD: (1% yeast extraxt, 2% peptone, 2% dextrose, 2% agar and see Supplementary Table 4 for composition of VSM and SSM) at 30 °C for 3 days. Some strains showed very poor growth on VSM, preventing measurement of colony size. These strains were not included in the figure. The tool Iris[71] was used for image analysis, resulting in tables of morphology scores and colony sizes (Supplementary Data 7; some data are missing for strain CEC5019, one of 35 cluster 13 strains) using an R script provided with Iris. For SSM results, Iris was unable to detect limits of colonies and we thus used ImageJ[72] to capture two perpendicular measures of diameter per colony. Analyses of variance (ANOVA) were performed using R, as well as post-ANOVA comparisons, Tukey's HSD (honest significant difference) tests[73]. Graphic representations (boxplots) were also performed using R.

**Check of premature stop codons by Sanger sequencing**. To confirm the presence of premature stop codons detected in silico in some ORFs, we Sanger sequenced ~400 bp regions within the *AFG1*, *BMT6*, *SFL1*, *VTA1*, and *ZCF29* ORFs, using DNA from 14 strains (two strains from cluster 1: SC5314 and CEC4496, two strains from cluster 2: CEC4493 and CEC4482, two strains from cluster 3: CEC3597 and CEC3681, two strains from cluster 4: CEC3536 and CEC3716, two strains from cluster 11: CEC3704 and CEC4525, four strains from cluster 13: CEC4103, CEC4104, CEC4878, CEC5030). Primer pairs were designed to amplify small regions of 400 bp using Primer3Plus[74] online (Supplementary Table 5). DNA was extracted using the extraction protocol in 96 deep-wells of the MasterPure™ Yeast DNA purification kit of epicenter. PCRs were performed in 50 μl reactions, using 0.5 μL Taq polymerase 5U (Thermofisher), 5 μL 10× buffer with KCl and without MgCl₂, 5 μL of dNTP 2 mM, 1 μl of each primer pair (10 μM) and 2 μL template DNA (concentration around 150 ng μL⁻¹ for all isolates). Amplifications were performed in a Mastercycler pro S from Eppendorf with a first denaturation step at 95 °C for 4 min, followed by 30 cycles of 40 s at 94 °C, 40 s at 55 °C and 40 s at 72 °C. The PCR program was followed by a final 10 min extension step at 72 °C. PCR products were purified and sequenced by the Eurofins Cochin Sequencing Platform in Paris, in one direction as sequences were short. Sanger sequences were verified by visual inspection.

**Code availability**. All code is available upon request to the authors.

**Data availability**. Raw reads have been deposited at the NCBI Sequence Read Archive under BioProject ID PRJNA432884 [https://www.ncbi.nlm.nih.gov/bioproject/432884]. Supplementary Data 2 and 3 are SNP datasets (see Methods section for filters). Supplementary Data 7 reports morphology sizes for colony growth on vaginal-simulative medium (VSM), saliva-simulative medium (SSM), and YPD at different temperatures (18 °C, 30 °C, 37 °C and 42 °C). All other relevant data are available from the corresponding author upon request.

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

## Acknowledgements

This work was supported by grants from the Agence Nationale de Recherche (ANR-10-LABX-62-IBEID), the Genoscope (projet #15 AP2008/2009 SNP *C. albicans*) and the Swiss National Science Foundation (Sinergia CRSII5_173863/1) to C.E., J.R. was supported by a Pasteur-Roux fellowship from Institut Pasteur. D.D. was the recipient of a PhD fellowship from Institut National de la Recherche Agronomique. E.P. was the recipient of a post-doctoral fellowship from the Wellcome Trust (WT088858MA). M.M.-H. and T.G. were supported by a grant from the Spanish Ministry of Economy and Competitiveness, BFU2015–67107 cofunded by the European Regional Development Fund (ERDF). C.E., M.-E.B., S.H.K., and J.H.S. were supported by a grant from the French and Korean Ministries for Foreign Affairs (PHC STAR 2011 25841YA). R.C.M. was supported by project MitoFun, funded by the European Research Council under the European Union's Seventh Framework Programme (FP/2007–2013)/ERC Grant Agreement No. 614562 and by a Wolfson Research Merit Award from the Royal Society. R.C.M. and K.V. were funded by the Surgical Reconstruction and Microbiology Research Centre, which is supported by the National Institute of Health Research, UK. G.S. was supported by the NIH grants R01-HG003468 and RO1-DE015873. C.E. and T.G. are members of the CNRS GDRI 0814 iGenolevures consortium. High-throughput sequencing has been performed on the Genomics Platform, member of France Géno-mique consortium (ANR10-INBS-09-08). We thank Bernard Dujon and Tatiana Giraud for providing insights on an earlier version of this manuscript.

## Author contributions

J.R., C.E., and M.-E.B. conceived and conducted the study. M.-E.B., K.V., R.C.M., A.M.B., A.C., S.F., S.H.K., P.L.P., O.R., and J.H.S. provided strains. N.S., K.M., C.Bo., L.M., K.S., K.V., R.C.M., J.P., P.W., C.Ba., and G.S. undertook wet-lab work and sequencing of the samples. J.R., C.M., D.D., M.M.-H., T.G., M.-E.B., A.P., E.P., and G.L. analyzed data. J.R. and C.E. wrote the manuscript and all authors revised the manuscript and made comments.

## Additional information

**Competing interests:** The authors declare no competing interests.

