## [Peer Review File · Nature Communications]

Reviewers' comments:

Reviewer #1 (Remarks to the Author):

The authors sequenced the genomes of 182 *Candida albicans* strains from a wide array of sources and analyzed them using a variety of population genetic and phylogenetic measures. Thanks to this massive effort, they identified five new strain clusters, in addition to confirming previously described clusters. The new clusters never contain more than three strains and thus could only have been found in such an extensive strain collection. Importantly, two of them (A and B) provide evidence for admixture, which they authors suggest could be due to *C. albicans*' parasexual cycle making this the first study providing evidence for the existence of the parasexual cycle in nature.

They furthermore show that one clade had significantly lower levels of heterozygosity than the others (clade 13 in Fig. 1). This clade also presents with the longest branch and the authors use it to root their ML tree (Fig. 2). While many clades in this tree are supported, some key branches lack support (backbone of lower half of tree). Could this be due to the rather small number of bootstrap replicates of 100 (p. 9)? Maybe this could be revisited.

If I understood the paper correctly, the authors root their ML tree with clade 13 due to its long branch, which can be done but may not reflect the true nature of the events. They could have used mid-point rooting instead. They then explain that *C. albicans*' sibling species *C. dubliensis* also has lower levels of heterozygosity (end of page 7, beginning page 8). But, how can two lineages with low or lower levels of heterozygosity give rise to one lineage with high levels of heterozygosity? Would a more parsimonious explanation not be the single gain of heterozygosity versus two losses of heterozygosity. Maybe the authors could clarify this statement.

Page 6: Could the authors clarify if all strains were measured for fitness in VSM?

Page 18: In the figure legend for the ML tree it states that 200 Bootstrap replicates were done but in the methods section it says 100.

Page 18: Figure legend for Fig. 3 – what are the asterisks specifying?

Reviewer #2 (Remarks to the Author):

This manuscript describes a sequence analysis of 182 strains of the human fungal pathogen *C. albicans* collected from a variety of sources around the globe. The authors have done a solid job of analyzing the relationships identified through sequence comparisons. These allow the authors to identify the fundamentally clonal patterns of *C. albicans* variation, but to characterize as well rare cases of sequence introgression they attribute to parasexual crosses – such crosses have been generated in the lab, but had not been previously noted in the wild.

Overall, the picture of an organism that is fundamentally clonal, with some limited evidence of genetic exchange, represents the perceived wisdom for *C. albicans* for many years now. The current manuscript adds resolution to the picture without changing the underlying framework. So what are the novelties that are identified that result from the high resolution available through the genomic sequencing?

First, the authors identify a level of chromosomal level aneuploidy that, while high, is not as high as

noted in some other studies – the idea is proposed that drug selections had generated a higher than normal level of aneuploidy in these earlier studies. So this work can provide a solid baseline to the inherent chromosomal aneuploidy in this organism. The data on admixture is also a useful addition to the field. However, the manuscript would be strengthened by some assessment of the potential mechanism generating the “parasexual” cross, as there seem to be challenges in getting strains sexually active even in laboratory conditions. A third new characterization is of the africana clade that shows clear genetic distinction from other clades, together with reduced virulence and slower growth. This variant appears to have frequent gene inactivation events that could impact virulence.

This paper thus provides an excellent example of the power as well as the limitations of large-scale genome sequencing efforts. It is primarily descriptive, allows a high-resolution analysis of patterns of genomic events like aneuploidy, inactivating mutations and recombination, but has less to say on mechanisms driving the observed events. It would provide an excellent baseline for directing experiments probing the mechanisms underlying these events.

Reviewers' comments:

Reviewer #1 (Remarks to the Author):

The authors sequenced the genomes of 182 *Candida albicans* strains from a wide array of sources and analyzed them using a variety of population genetic and phylogenetic measures. Thanks to this massive effort, they identified five new strain clusters, in addition to confirming previously described clusters. The new clusters never contain more than three strains and thus could only have been found in such an extensive strain collection. Importantly, two of them (A and B) provide evidence for admixture, which they authors suggest could be due to *C. albicans*' parasexual cycle making this the first study providing evidence for the existence of the parasexual cycle in nature.

They furthermore show that one clade had significantly lower levels of heterozygosity than the others (clade 13 in Fig. 1). This clade also presents with the longest branch and the authors use it to root their ML tree (Fig. 2). While many clades in this tree are supported, some key branches lack support (backbone of lower half of tree). Could this be due to the rather small number of bootstrap replicates of 100 (p. 9)? Maybe this could be revisited.

If I understood the paper correctly, the authors root their ML tree with clade 13 due to its long branch, which can be done but may not reflect the true nature of the evens. They could have used mid-point rooting instead.

Reply: We thank the reviewer for this comment. According to this recommendation, we have now performed 1,000 bootstrap replicates and used the midpoint rooting method to root our tree. As shown in revised Fig. 2, the topology of the tree has not changed upon midpoint rooting and increasing the number of bootstraps did not allow providing support to those branches that lacked support in the previous analysis.

They then explain that *C. albicans*' sibling species *C. dubliniensis* also has lower levels of heterozygosity (end of page 7, beginning page 8). But, how can two lineages with low or lower levels of heterozygosity give rise to one lineage with high levels of heterozygosity? Would a more parsimonious explanation not be the single gain of heterozygosity versus two losses of heterozygosity. Maybe the authors could clarify this statement.

Reply: We thank the reviewer for this comment and have modified the manuscript to explain why we favor a hypothesis where Cluster 13 and *C. dubliniensis* have arisen by loss-of-heterozygosity from a heterozygous ancestor. The manuscript now reads "Cluster 13 isolates however have much lower heterozygosity compared to other strains of *C. albicans* (Fig. 1). This may reflect a combination of massive ancestral LOH events and clonal reproduction in this cluster, with fixation of several deleterious alleles, affecting the overall fitness of these strains and leading to its niche restriction. Notably, the closest relative of *C. albicans*, namely *Candida dubliniensis*, also shows lower

heterozygosity⁴⁵. It has been reported that *C. dubliniensis* is less virulent and has lower fitness compared to *C. albicans*^{46–49}, and that its genome harbors numerous pseudogenes and a lower level of genetic diversity. In the CTG clade of *Saccharomycotina* to which *C. albicans* and *C. dubliniensis* belong, more distantly related diploid species show high levels of heterozygosity similar to (or even higher than) those observed for non-cluster 13 isolates of *C. albicans*. These species have been shown to result from hybridization events^{50–52}. Cluster 13 isolates of *C. albicans* showed ancient LOH in telomere-proximal regions, suggesting these events have occurred by break-induced replication (BIR) and/or mitotic recombination, as regions around centromeres showed higher heterozygosity (Fig. 1 and Supplementary Fig. 7). This pattern of heterozygosity is different from what we observed in other *C. albicans* isolates, in which heterozygosity was steady along chromosomes, except when recent LOH were observed (Fig. 1 and Supplementary Fig. 7). The most parsimonious ancestral state is thus a high level of heterozygosity, followed by two independent events (homoplasmy) of massive losses of heterozygosity and accumulations of deleterious alleles in *C. dubliniensis* and isolates of cluster 13 of *C. albicans*.”

Page 6: Could the authors clarify if all strains were measured for fitness in VSM?

Reply: We apologize not to have been clearer in our manuscript. We tried to measure fitness in VSM for all strains but some strains showed very poor growth on that medium, preventing measurement of colony size. These strains were not included in the figure. We have clarified the method.

Page 18: In the figure legend for the ML tree it states that 200 Bootstrap replicates were done but in the methods section it says 100.

Reply: Again, we apologize to have done this mistake. We made 200 bootstrap replicates. In the revised manuscript, thanks to the first comment of the reviewer, we performed 1,000 bootstrap replicates and have modified figure 2 and the manuscript accordingly.

Page 18: Figure legend for Fig. 3 – what are the asterisks specifying?

Reply: We apologize for this oversight. Asterisks represent strains that could not be assigned to any genetic clusters. We completed the figure legend accordingly.

Reviewer #2 (Remarks to the Author):

This manuscript describes a sequence analysis of 182 strains of the human fungal pathogen *C. albicans* collected from a variety of sources around the globe. The authors have done a solid job of analyzing the relationships identified through sequence comparisons. These allow the authors to identify the fundamentally clonal patterns of *C. albicans* variation, but to characterize as well rare cases of sequence introgression they attribute to parasexual crosses – such crosses have been generated in the

lab, but had not been previously noted in the wild.

Overall, the picture of an organism that is fundamentally clonal, with some limited evidence of genetic exchange, represents the perceived wisdom for *C. albicans* for many years now. The current manuscript adds resolution to the picture without changing the underlying framework. So what are the novelties that are identified that result from the high resolution available through the genomic sequencing?

First, the authors identify a level of chromosomal level aneuploidy that, while high, is not as high as noted in some other studies – the idea is proposed that drug selections had generated a higher than normal level of aneuploidy in these earlier studies. So this work can provide a solid baseline to the inherent chromosomal aneuploidy in this organism. The data on admixture is also a useful addition to the field. However, the manuscript would be strengthened by some assessment of the potential mechanism generating the “parasexual” cross, as there seem to be challenges in getting strains sexually active even in laboratory conditions. A third new characterization is of the *africana* clade that shows clear genetic distinction from other clades, together with reduced virulence and slower growth. This variant appears to have frequent gene inactivation events that could impact virulence.

This paper thus provides an excellent example of the power as well as the limitations of large-scale genome sequencing efforts. It is primarily descriptive, allows a high-resolution analysis of patterns of genomic events like aneuploidy, inactivating mutations and recombination, but has less to say on mechanisms driving the observed events. It would provide an excellent baseline for directing experiments probing the mechanisms underlying these events.

Reply: We thank the reviewer for these positive comments on our manuscript. We have added a paragraph discussing mating and parasexuality in *C. albicans*: “Genetic differentiation occurs when gene flow is prevented, due to reproductive barriers or asexuality. Genes involved in mating were shown to be under purifying selection in *C. albicans*²⁸, suggesting that they are still functional. Accordingly, mating has been induced between various *C. albicans* isolates of opposite mating-types as well as between *C. albicans* and its close relative *C. dubliniensis*, in the laboratory in vitro and in vivo, leading to the formation of tetraploids^{29–32}. Return to the diploid state has also been observed and involved random chromosome loss rather than meiosis⁵. While these data suggest that genetic differentiation in the *C. albicans* species does not impose prezygotic barriers, these experiments did not systematically address possible genetic incompatibilities between genetic clusters. In particular, they did not assess the existence of postzygotic reproductive isolation, i.e. non-viability or sterility of hybrids. Thus a thorough investigation of reproductive isolation between different genetic clusters in *C. albicans* is still lacking. Importantly, the two newly identified genetic clusters showing footprints of admixture were unknown. Our resource of 182 genome-sequenced isolates will be invaluable to address this key question.”

REVIEWERS' COMMENTS:

Reviewer #1 (Remarks to the Author):

All previous comments were addressed sufficiently.